# Transvaginal Natural Orifice Transluminal Endoscopic Surgery (vNOTES) in Urogynecological Surgery: A Systematic Review

**DOI:** 10.3390/jcm13195707

**Published:** 2024-09-25

**Authors:** Lorenzo Vacca, Eleonora Rosato, Riccardo Lombardo, Paolo Geretto, Simone Albisinni, Riccardo Campi, Sabrina De Cillis, Laura Pelizzari, Maria Lucia Gallo, Gianluca Sampogna, Andrea Lombisani, Giuseppe Campagna, Alessandro Giammo, Vincenzo Li Marzi, Cosimo De Nunzio

**Affiliations:** 1Gynecological Surgery Unit, Dipartimento Centro di Eccellenza Donna e Bambino Nascente, Ospedale Isola Tiberina—Gemelli Isola, 00136 Rome, Italy; lorenzovacca08@gmail.com (L.V.); andrea.lombisani@gmail.com (A.L.); giuseppe.campagna@gmail.com (G.C.); 2Unit of Urology, Department of Surgical Sciences, Tor Vergata University Hospital, Tor Vergata University of Rome, 00133 Rome, Italy; eleonoraros92@gmail.com (E.R.); albisinni.simone@gmail.com (S.A.); 3Unit of Urology, Sant’Andrea Hospital, Sapienza University, 00189 Rome, Italy; cosimodenunzio@virgilio.it; 4Unit of Neuro-Urology, Città della Salute e della Scienza University Hospital, University of Turin, 10126 Turin, Italy; paolo.gere@gmail.com (P.G.); giammo.alessandro@gmail.com (A.G.); 5Department of Minimally Invasive and Robotic Urologic Surgery, Careggi University Hospital, University of Florence, 50134 Florence, Italy; riccardo.campi@gmail.com (R.C.); marialucia.gallo@unifi.it (M.L.G.); 6Division of Urology, Department of Oncology, San Luigi Gonzaga Hospital, University of Turin, 10043 Turin, Italy; sabrinatitti.decillis@gmail.com; 7Department of Rehabilitative Medicine, AUSL Piacenza, 29121 Piacenza, Italy; laura.pelizzari.23@gmail.com; 8Unit of Urology, Niguarda Hospital, University of Milan, 20162 Milan, Italy; 9Department of Medical, Surgical and Neurological Science, University of Siena, 53100 Siena, Italy; vlimarzi@hotmail.com

**Keywords:** vNOTES, minimally invasive, pelvic organ prolapse

## Abstract

**Background:** Minimally invasive surgery could improve cosmetic outcomes and reduce the risks of surgical injury with less postoperative pain and a quicker patient’s discharge. Recently, transvaginal natural orifice transluminal endoscopic surgery (vNOTES) has been introduced in urogynecology with exciting results. **Evidence Acquisition:** After PROSPERO registration (n°CRD42023406815), we performed a comprehensive literature search on Pubmed, Embase, and Cochrane CENTRAL, including peer-reviewed studies evaluating transvaginal natural orifice transluminal endoscopic surgery. No limits on time or type of study were applied. **Evidence synthesis:** Overall, 12 manuscripts were included in the analysis. Seven studies evaluated uterosacral ligament suspension, four studies evaluated sacral colpopexy, three evaluated sacrospinous ligament suspension, and one study evaluated lateral suspension. Overall success rates were high (>90%); however, definitions of success were heterogeneous. In terms of complication, most of the studies reported low-grade complications (Clavien–Dindo I and II); only two patients needed mesh removal because of mesh exposure. The risk of bias of the trials was rated in the medium to high-risk category. **Conclusions:** The present review highlights important initial results for vNOTES. Future randomized clinical trials are needed to better define its role in the management of urogynecological procedures.

## 1. Introduction

In order to reduce surgical morbidity, advances in endoscopic and optic technologies allowed the development of a less invasive alternative to conventional laparotomy and laparoscopic surgery [1]. Between these surgical alternatives, single-incision laparoscopic surgery (SILS) and natural orifice transluminal endoscopic surgery (NOTES^®^) have gained popularity among general surgeons, gynecologists, urologists, and gastroenterologists over the past few years [2], and their feasibility and safety have been approved [1,3]. 

While vNOTES has been around for over a decade, most obstetric and gynecologic (OB/GYN) surgeons remain unfamiliar with the procedure. As multiport laparoscopy and robotics began replacing open surgery in various specialties, the search for further innovations led to the development of laparoendoscopic single-site surgery (LESS). LESS involves passing multiple instruments, typically a camera and two or three additional tools, through a single port, usually positioned at the umbilicus. This method offers benefits like improved cosmetic results and potentially less pain [4,5].

Several commercially available LESS port options exist, but in low-resource settings, particularly in developing countries where these commercial ports may not be accessible, surgeons sometimes resort to creating a self-made port. This is performed by placing a surgical glove over a small wound retractor, though it is important to note that this off-label use has not received Food and Drug Administration (FDA) approval. NOTES evolved as an extension of LESS, aiming to access the abdominal cavity through natural orifices such as the stomach, esophagus, rectum, bladder, and vagina. Among these approaches, transvaginal vNOTES quickly became the preferred method due to its lower risk of contamination and better visualization, closure, and healing. Initially, early adopters of vNOTES used a self-constructed LESS port, which involved fitting a sterile glove over a wound retractor and inserting ports into the glove fingers. However, in 2019, the FDA approved a commercial product specifically designed for vNOTES, the GelPOINTR V-Path transvaginal access platform by Applied Medical in Rancho Santa Margarita, California [4,5].

Minimally invasive surgery could improve cosmetic outcomes and reduce the risks of surgical injury with less postoperative pain and a quicker patient discharge. In particular, NOTES uses natural orifices as surgical access and can be performed via a variety of approaches, including the stomach, esophagus, bladder, and rectum [2,3]. However, the majority of NOTES procedures have been performed using vaginal approaches (vNOTES). vNOTES can avoid incision scars on the abdominal wall and ensure easier access in obese patients (moderate Trandelenburg position is required) or in patients with previous abdominal surgery (presence of midline laparotomy or mesh hernia repairs) [4,5].

From a technical perspective, vNOTES offers several advantages over transabdominal LESS, such as a larger and more flexible colpotomy opening, reduced instrument clashing, and better visualization due to the “bottom-up” camera positioning and smoke plume dispersion. The vNOTES approach also reduces the need for extensive adhesiolysis in patients with severe abdominal adhesions and provides better maneuverability and control of the uterine blood supply during procedures for enlarged uteri. In pelvic organ prolapse (POP) surgery, vNOTES offers improved visualization for precise suturing, easier adnexectomy, and significant cosmetic benefits due to the absence of visible laparoscopic scars.

The first published experience was with transvaginal endoscopic cholecystectomy performed by Zorron et al. [6], but more recently, this approach has been used for appendectomy, nephrectomy, and especially gynecological procedures [1].

In gynecology, the vagina is considered a true additional route for surgery [7]. In comparison with conventional transvaginal surgery, the surgical field of transvaginal NOTES can be demonstrated clearly with endoscopic guidance with the assistance of laparoscopic instruments [8]. In the beginning, it was used only for diagnostic purposes or performed with transabdominal assistance; now, it is being used for complex procedures [1]. Indeed, vNOTES may improve visualization for complex pelvic surgery compared to traditional vaginal approaches [8]. The main limitation is represented by the spatial conflict between instruments. Transvaginal NOTES is especially beneficial in patients with virginity, nulliparity, obesity, or a narrow vagina, which are all considered relative contraindications in conventional transvaginal surgery [9].

In gynecology, vNOTES first emerged in 2012 [1]. The initial published series were based on patients treated for adnexal disease, hysterectomy for benign disease, myomectomy, and early-stage endometrial cancer [1].

Recently, vNOTES has also been used to perform urogynecology procedures such as sacral colpopexy (SCP) and uterosacral ligament suspension (USLS) [8], showing its safety and feasibility. However, only a few articles were published. Therefore, it is clinically relevant to thoroughly research the safety and efficacy of this procedure, given the paucity of existing data [7].

The primary aim of this systematic review is to summarize the surgical aspects and functional outcomes present in the literature on the use of vNOTES in urogynecology procedures.

## 2. Evidence Acquisition

The present systematic review was performed and reported according to the Preferred Reporting Items for Systematic Reviews and Meta-analyses (PRISMA) statement [10]. 

### 2.1. Information Sources, Search Strategy, and Selection Process

After PROSPERO registration (n°CRD42023406815), we performed a comprehensive literature search on Pubmed, Embase, Google Scholar, Web of Science, and Cochrane CENTRAL, including peer-reviewed studies. The literature search was performed in October 2023 by L.V., E.R., and R.L. The keyword search was performed using both Medical Subject Headings (MeSH) terms and free text. The research strategy protocol is depicted in Appendix A and was approved by PROSPERO on the 20 March 2023.

### 2.2. Inclusion and Exclusion Criteria

Two authors (L.V. and S.A.) independently screened all titles, abstracts, and full-text records against the eligibility criteria by collecting them in an Excel sheet. In case of any disagreement, a third author (R.L.) solved the divergence. No automation tools were used. 

Studies were selected using the following PICO protocol. Patients: women undergoing urogynecology surgery; Intervention: vNOTES; Comparator: any other urogynecology intervention; Outcome: success. We included cohort and case–control studies, as well as non-randomized (NRCT) or randomized trials (RCT). 

Exclusion criteria included. Manuscripts reporting <10 cases or written in languages other than English were excluded, as were review articles, video articles, or articles not inherent to the topic.

### 2.3. Data Collection Process and Data Items

Eligible outcomes were broadly categorized as follows: (1) baseline characteristics; (2) type of surgery and peri- and postoperative outcomes; (3) pre- and post-clinical evaluations and quality of life questionnaires.

We collected information about the authors, years of publication, and the institution where the study has been conducted. The design of the study was also reported.

The number of participants in each included study, as well as the age of the participants (mean ± SD, range, and median), postmenopausal status (number or rate), and parity (number, rate, mean, and/or median) were extracted. Numbers and rates of previous hysterectomy, pelvic surgery, and pelvic organ prolapse (POP) surgery were reported. 

For surgical aspects, the following items were extracted: type of surgery, number of patients or rates of concomitant anti-incontinence procedures, anterior or posterior colporrhaphy, and hysterectomy. We collected data on operative time (minutes, mean ± SD/range, median), blood loss (mL, mean ± SD/range, median), and, finally, conversion rates and hospitalization stay (days, mean ± SD/range, median).

For functional evaluations, we reported pre- and postoperative pelvic organ prolapse quantification system (POP-Q, mean ± SD/range, median, number or rate), de novo stress urinary incontinence (SUI, number, rate), de novo constipation (number, rate), de novo dyspareunia (number, rate), anatomical success rate and its definition, quality of life (QoL) questionnaires (type and postoperative scores), POP recurrence or reoperation rates, and months of follow-up (mean ± SD/median and range). 

Data of selected studies are summarized in Table 1, Table 2, Table 3 and Table 4 [4,7,8,9,11,12,13,14,15,16,17,18]. 

### 2.4. Reporting Bias Assessment

The risk of bias of each included study was assessed by two review authors working independently using the Cochrane Collaboration Risk of Bias Tool for Randomized Controlled Trials (Rob 2) and the Risk of Bias In Non-randomised Studies of Interventions (ROBINS-1) scale for randomized and non-randomized observational studies, respectively (Table 5). 

## 3. Evidence Synthesis

### 3.1. Study Selection

Following an initial search, a total of 602 publications were identified through database searching. After an initial screening, 567 studies were excluded by title, abstracts, duplication, and other languages but English. Thirty-five studies were assessed for eligibility. Finally, 12 manuscripts were included (Table 1). Figure 1 provides a diagram of the flow of information through the different phases of this systematic review according to the PRISMA criteria [10]. 

Overall, six studies were retrospective cohort studies [9,10,11,12,13,14,15], while six were prospective studies [4,7,16,17,18,19]. No randomized clinical trials were retrieved.

### 3.2. Study Characteristics 

The overall number of patients was 414, with a minimum of 15 [17] and a maximum of 65 [12]. Eleven studies reported data about the age of patients, with a range of 40–81 years. Information about postmenopausal state was available in only 3/12 studies [13,16,18]. Data concerning previous pelvic surgery were available in 5 studies, and 18% (32) of the patients included in these studies underwent it [7,11,17,19,20]. A very low rate (<5% of patients) has undergone a previous POP surgery. In terms of surgical procedure types, 5 (41.7%) studies evaluated uterosacral ligament suspension (USLS, 181 patients) [4,12,14,17,20], and one study including 15 patients evaluated USLS associated with anterior longitudinal ligament suspension (ALLS) [18]. Two studies evaluated sacrocolpopexy (SCP, 81 patients) [9,12], 2 evaluated sacrospinous ligament suspension (SSLS, 49 patients) [16,19], and one study evaluated lateral suspension (LS, 37 patients) [7]. Finally, one study compared SCP (12 patients), USLS (38 patients), and SSLS (1 patient) [14] performed in a series of vNOTES patients. 

### 3.3. Surgical Outcomes 

#### 3.3.1. Intra- and Perioperative Data 

Overall, most of the interventions included concomitant hysterectomy (>90%) and concomitant posterior colporrhaphy (>30%). Finally, one hundred twenty patients underwent anterior colporrhaphy. Quite the opposite, a concomitant anti-incontinence procedure such as mid-urethral sling positioning was performed in less than 20% of the cases [7,11,12,13,19].

The mean operative time ranged, respectively, from 115 to 189 min for SCP (REF. Lou e Liu), from 52 to 141 min for USLS (REF Huang 2022 e Ahroni), and from 63 to 192 min for SSLS surgery [15,18]. Estimated blood loss (EBL) was similar among the different techniques (30–147 mL for SCP vs. 30–100 mL for USLS vs. 82–103 mL for SSLS) [8,15,18,20]. Generally, the conversion rate was low, with a range from 0 to 12% for SCP [13,20] and a single case of conversion to vaginal surgery for USLS [4]. No conversion was needed for SSLS [14,15,18]. The higher mean hospital stay was reported for SSLS procedures (4–8 days), followed by USLS (2–7 days) [16,19] and SCP (4–5 days) [12,14,20] (Table 2 and Table 3).

#### 3.3.2. Anatomical and Functional Outcomes

Overall, most of the included studies (11 out of 12) used the POP-Q system to evaluate anatomical outcomes. Additionally, anatomical success rates and various patient-reported outcomes were assessed using dedicated questionnaires (PFIQ, FSFI, PFDI, POPDI, PISQ). Finally, recurrence and reoperation rates were also reported. Table 3 provides a detailed description of the anatomical and functional outcomes for each study.

Overall success rates were evaluated using the pre- and postoperative POP-Q system, as were quality of life and reoperation rates. Overall success rates were high, over 90%; however, their definitions were heterogeneous [12,13,15]. Quality of life was evaluated using different questionnaires (Table 3); all studies showed improvements in terms of quality of life after surgery. Although it is a very important aspect for the success evaluation, de novo stress urinary incontinence (SUI), constipation, or dyspareunia were scarcely reported apart from the Lu et al. study [13]. 

Reoperation rates were low, ranging from 0 to 25% in the Huang et al. series [15]; however, follow-up was short in most of the selected studies, with a minimum of 6 weeks [4], and only a single study showing a long mean follow-up of 35 ± 7.6 months [12].

#### 3.3.3. Complications 

In the included studies, intraoperative and postoperative complications were scarce and of low grade, considering Satava Classification [21] and Clavien–Dindo Classification [22], respectively. Intraoperative complications were reported in three studies [4,7,11]. In particular, Aharoni et al. [11] reported a single case of intra-abdominal bleeding (Grade I), a single case of vaginal bleeding (Grade II), and two cases of cystotomy (Grade II). Bladder injury was reported in 8.1% of the Ketenci et al. [7] population, as well as a single case of epigastric vessel injuries. A single case of ureteral kinking occurred in the Farah et al. series [4].

Postoperative complications are summarized in Table 4 and were more frequent. Urinary retention was the most frequent Grade I complication according to Clavien–Dindo Classification (5 cases) [4,12,13], while a single case of ileus conservatively treated (Grade I) was reported [20]. Infections, urinary tract infections, and vaginitis were possible complications (Grade II), presented in 9 (2.2%) cases in total [7,12,13,23]. Bleeding was rare (3/414 patients) [7,11,12] and always treated conservatively (hematoma, Grade II). Five (1.2%) patients showed pain at the site of insertion or pelvic pain (Grade II) [7,18]. Considering vNOTES mesh POP surgeries [7,12,13,14], the most worrying complication, mesh erosion, was reported in 3/130 (2.3%) patients; one case required estrogen treatment (Grade II) [7], while the others required surgical removal (Grade IIIb) [7,15]. 

## 4. Discussion 

This systematic review was designed to investigate the efficacy, safety, and feasibility of vNOTES for POP surgery. First described in 2012, vNOTES appears to combine the advantages of transvaginal procedures with laparoendoscopic single-site surgery, in addition to better cosmetic outcomes [24].

Concerning perioperative data, surgery time was comparable to those reported by other groups for the performance of conventional vaginal SSLF and USLS [25] and of minimally invasive abdominal procedures, including SCP [26], USLS [27], and LS [28]. 

Even though most of the vNOTES POP reconstructive procedures were associated with hysterectomy, overall estimated blood loss (EBL) was superior to that registered after a conventional laparoscopic approach [26,27,28], but when compared with conventional vaginal procedures, vNOTES appeared to be safer in terms of blood loss. This may be related to the enhanced visualization of anatomical structures, allowing an easier approach to the infundibulopelvic and ovarian ligaments [8]. Moreover, even though vaginal hysterectomy is the preferred route of hysterectomy in non-oncological disease, a vaginal salpingo-oophorectomy could sometimes be technically challenging and could need to be converted to an abdominal approach, even in the case of the most skilled surgeon [29,30]. 

Additionally, considering the evidence that ovarian serous carcinoma may originate in the distal fallopian tube with the consequent protective effect of prophylactic salpingectomy [31,32] in preventing ovarian cancer at the time of hysterectomy (as recommended by a position paper published by the American College of Obstetrics and Gynaecology (ACOG) in April 2019) [33], it is even more evident the advantage in terms of operative times and better cosmetic outcomes, avoiding a combined approach [34,35]. 

Concerning the overall perioperative complication rate, they seemed to be low and similar to the conventional laparoscopic and vaginal approaches [25]. Similar to the L-USLS, vNOTES USLS demonstrate a minor incidence of ureteral kinking or damage (only one case) in comparison to the traditional vaginal approach. This is related to the possibility of performing anatomical dissection and suturing under direct visualization of the ureter, similarly for SSL, reducing the risk of rectal damage and neurovascular complications. In terms of infection control, the selected studies applied different strategies. More specifically, antibiotic prophylaxis with a cephalosporin alone or in combination with metronidazole was performed preoperatively. In the case of postoperative antibiotic prophylaxis, the cephalosporin was stopped within 48 h after the surgery was completed. Overall, the rate of infections using these strategies was low (2.2%). 

Not least, the risk of mesh exposure in the case of SCP was higher in comparison to laparoscopic and robotic approaches [36]. This may be related to the fact that, in these cases, a total hysterectomy was performed. It is undeniable that the cervix must be preserved to reduce the risk of mesh exposure [37,38].

In conclusion, the reduced operative time combined with the low EBL and perioperative complication rate makes this kind of surgery suitable for elderly women with associated comorbidities, avoiding the Trendelenburg position.

Anatomical outcomes were evaluated using the POP-Q system, suggesting, when specified, a postoperative POP-Q stage < 2 as a success criterion. The overall success rate was over 90% in almost all the studies included in our report. Even when anatomical success criteria were not specified, the low rate of POP recurrence and reoperation rate underlined the efficacy of the technique. Probably, the direct vision of UL and SSL during vNOTES surgeries may improve the capacity of surgeons to suture the ligaments in a more effective and precise way through direct vision, which differs from the conventional vaginal approach. The short follow-up period in most of the studies should be taken into account in the interpretation of the data, considering the recent introduction of the vNOTES technique in urogynecological surgical practice. However, according to questionnaire data, pelvic floor reconstructive surgery with vNOTES appears to be effective in improving urinary, colorectal, and sexual symptoms that negatively affect the quality of life of women with POP.

The learning curve of vNOTES surgery has been evaluated by some authors. According to the international consensus among vNOTES experts, it is recommended that beginners start by learning how to perform a complete hysterectomy via vNOTES [39,40]. This is because the surgical pathway created during this procedure is more accessible than the culdotomy approach through the posterior or anterior vaginal fornix. Others argue that vNOTES has a steep learning curve and can be particularly challenging for those without experience.

It is important to recognize that vNOTES is a relatively new and evolving procedure, with its indications and contraindications being continuously updated as new clinical evidence emerges and technological advancements are made. According to the available literature on the learning curve in vNOTES surgery, expert surgeons may need less than 10 cases to achieve the learning curve; however, young surgeons may need more time to master the technique. 

vNOTES also presents unique challenges, particularly the need for the surgeon to perform both anterior and posterior colpotomies, which has emerged as a significant barrier to the widespread adoption of this technique. These colpotomies become even more challenging in complex cases involving difficult vaginal access, such as a narrowed introitus and vaginal canal, limited descent, a narrow pubic arch, or conditions like obesity, nulliparity, postmenopausal atrophy, and testosterone use in transgender patients. Addressing these obstacles requires a high level of skill [41].

One potential solution is for urogynecologists and vaginal surgeons to collaborate with surgeons who are mastering vNOTES, allowing experienced surgeon to guide their colleagues through the colpotomies. Notably, in a large series of over 1000 hysterectomy cases performed by high-volume surgeons, cystotomy rates were low at 1.2% [42].

Additionally, surgeons need to reconsider their approach to case selection. For instance, endometriosis excision is generally better performed using a transabdominal laparoscopy or robotic approach. While the “bottom-up view” in vNOTES offers certain advantages, it can also be limiting, as it may prevent the surgeon from seeing potential pathology, such as middle and upper abdominal bowel adhesions to a large fibroid uterus. This limitation may necessitate a lower threshold or higher conversion rate to laparoscopy. Although conversions are typically metrics to avoid, transitioning from vNOTES to a laparoscopic approach is often seen as reverting to a standard technique [26].

The role of preoperative imaging in vNOTES remains unclear, though high-quality transvaginal ultrasound or magnetic resonance imaging may be beneficial in select cases.

From a technical perspective, vNOTES offers several advantages over transabdominal LESS. These include a larger and more flexible opening following vNOTES colpotomy compared to minilaparotomy or transabdominal incisions. Additionally, the proximity of the target tissue to the port reduces restricted movements and minimizes instrument clashing. The uterus also serves as an additional triangulation point, positioned closer to both the instruments and the scope. Moreover, the smoke plume disperses into the upper abdomen, reducing the likelihood of obstructing the view. The use of bariatric scopes, a 30-degree lens, and an angled adapter for the light source further decreases the chances of camera and instrument collisions, often referred to as “sword fighting” [40]. Another potential benefit unique to vNOTES is the reduced need for surgical assistants. In this approach, the assistant primarily handles the camera while the primary surgeon operates independently, akin to robotic surgery, but without the requirement for a highly skilled and costly bedside assistant. Additionally, the “bottom-up” approach offers superior visualization, as the camera is positioned much closer to the target tissues, such as the ureter at the level of the infundibular ligaments or the uterine arteries and uterosacral ligaments during apical suspension.

Another example involves patients with severe mid and upper abdominal adhesions from prior surgeries, where the vNOTES approach may eliminate the need for extensive adhesiolysis, which would be necessary with transabdominal laparoscopic or robotic approaches. Additionally, when performing hysterectomies for enlarged uteri, such as those with large fibroids or adenomyosis, the vNOTES technique allows the surgeon to move the uterus cephalad, further into the abdomen, rather than deeper into the pelvis as in transabdominal laparoscopy. This creates more space for the operation and enables control of the uterine blood supply at the outset of the procedure [43].

Regarding POP surgery more specifically, when apical suspension is required, vNOTES offers the potential for enhanced visualization and more precise placement of higher apical sutures, which may reduce the risk of ureteral kinking. Although there is a lack of specific data on ureteral injury and kinking rates with vNOTES uterosacral ligament suspensions, existing studies on vaginal and laparoscopic approaches suggest that the surgical route may influence these outcomes. Additionally, adnexectomy may be performed more easily with vNOTES compared to traditional transvaginal surgery, particularly in cases like bilateral salpingo-oophorectomy for breast cancer or transgender patients. The absence of visible laparoscopy scars is a significant cosmetic benefit for transgender patients undergoing hysterectomies, as well as for other patients who wish to conceal salpingectomies for sterilization or hysterectomies for personal or cultural reasons [26,41,42,43].

As awareness of work-related musculoskeletal disorders (WRMDs) and workplace injuries grows, with a focus on prevention, early recognition, treatment, and research, vNOTES offers a potential ergonomic advantage. Unlike laparoscopy and vaginal surgery, where surgeons are often required to remain in non-neutral, constrained positions for extended periods, vNOTES allows the surgeon to operate in a more neutral position, either seated or standing, without the need for prolonged static posture at a robotic console. Furthermore, due to improved surgeon posture and movement, the mechanics and ergonomics of vNOTES are potentially more favorable compared to unassisted transvaginal surgery.

In the past few years, some authors have introduced the use of robotic platforms to perform vNOTES surgery; however, for the time being, there are only some case series available in the literature. The introduction of the single-port Da Vinci surgical system is being tested by a single institution and may represent a useful tool for the vNOTES approach [43,44]. 

Since urogynecologists are the primary instructors of vaginal hysterectomy for residents and fellows, a key benefit of converting vaginal uterosacral ligament suspensions to vNOTES, due to its enhanced technique, would be the opportunity to train the next generation of surgeons in this approach. These new surgeons, trained in vNOTES, are more likely to incorporate the technique into their practices and, in turn, pass on their knowledge to others. As vNOTES becomes more widely adopted, procedures such as salpingectomies for sterilization, prophylactic bilateral salpingo-oophorectomies, uterosacral ligament hysteropexies, and transgender hysterectomies are likely to shift from laparoscopic or robotic methods to the vNOTES approach [45,46].

The rise of laparoscopy marked the beginning of a decline in vaginal surgery, with the advent of robotics further reducing its prevalence. Despite the higher costs and steep learning curve associated with laparoscopic surgery, it became increasingly popular due to enhanced visualization and the development of techniques that were not feasible through open or vaginal approaches. As a result, some surgeons transitioned away from vaginal surgery in favor of laparoscopic and robotic methods, leaving newly trained fellows and residents with limited exposure to vaginal procedures [47,48].

vNOTES could be a pivotal technique for the next generation of surgeons, who are already accustomed to laparoscopic methods and may be more inclined to adopt it. For older surgeons who may feel excluded from the advancements in robotics and laparoscopy, vNOTES offers an opportunity to acquire new laparoscopic skills while also preserving and passing on their expertise in vaginal surgery. The shared objective would be to build a supportive community of practice, providing both immediate and long-term guidance. Surgical coaching models could play a valuable role in this effort. We anticipate and hope that vNOTES will help revive vaginal surgery, paving the way for future innovations and advancements in the field [48,49,50].

Finally, the landscape of surgical innovation is being transformed by the recent advancements in AI and natural language processing (NLP). The integration of AI into the management of patients with pelvic organ prolapse (POP) has the potential to enhance patient selection, identifying those who would benefit most from POP surgery or are suitable candidates for the vNOTES approach. Additionally, AI-integrated systems can improve the surgical skills of clinicians, leading to better surgical outcomes. The introduction of NLP models, such as chatbots, could further enhance patient information and reporting by assisting clinicians in communicating effectively with patients, thereby improving patient-reported outcomes [51,52,53,54]. The integration of diagnostic imaging into laparoscopic and robotic systems may help surgeons identify anatomical landmarks such as ureters and vascular structures. This may lower the complication rates and improve surgical outcomes. Overall, AI and NLP are crucial tools that hold great promise for advancing the future of vNOTES surgery.

The limitations of this review include the variation in definitions of success across the included studies, making direct comparisons of outcomes challenging. The risk of bias in the studies was rated as medium to high, potentially affecting the reliability of the findings. Additionally, most studies had short follow-up periods, limiting the assessment of long-term outcomes and complications. The review did not include any randomized controlled trials, which are the gold standard in clinical research. Furthermore, the steep learning curve associated with vNOTES may hinder its broader adoption in clinical practice. The absence of restrictions on time and study type may be considered a limitation of this study. However, due to the limited number of studies available on the subject, we chose to take a broader approach to provide a comprehensive overview of the current state of the art of the technique. However, notwithstanding these limitations, the present review provides a clear picture of vNOTES surgery nowadays.

## 5. Conclusions

According to our systematic review, vNOTES surgery appears to be a safe and effective approach for patients with pelvic organ prolapse. Moreover, the results identify USLS as a safe and effective technique for the suspension of the apex in POP. However, further prospective randomized studies with a longer follow-up are needed to confirm the available data. Clinicians beginning to perform vNOTES surgery should proceed carefully, ensuring adequate mentorship and careful case selection to maximize outcomes. Meanwhile, clinicians who are experts in vNOTES should focus on designing well-structured clinical trials to better define the role of vNOTES in urogynecological surgery.

## Figures and Tables

**Figure 1 jcm-13-05707-f001:**
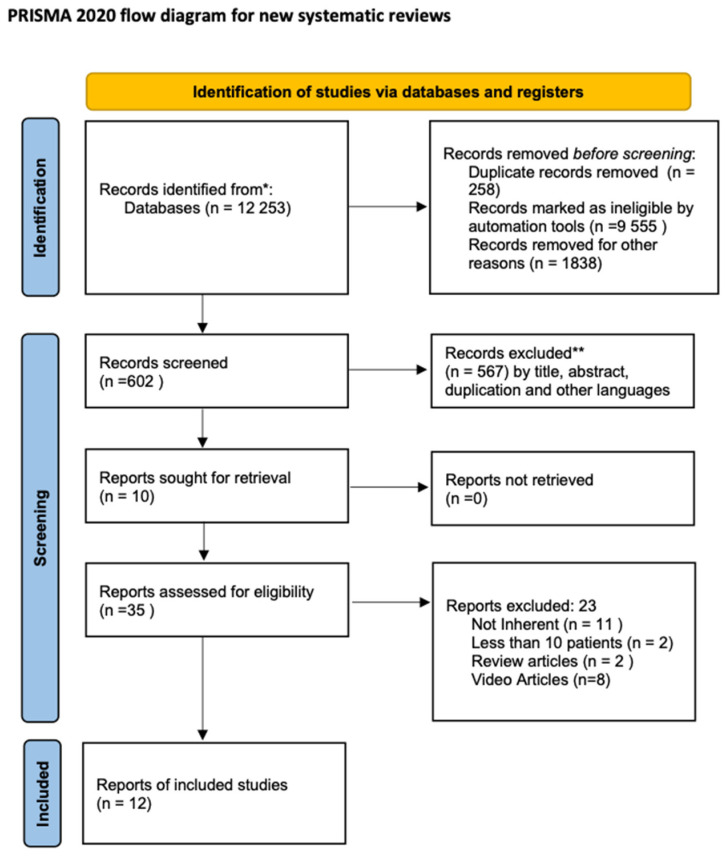
PRISMA flow chart. * Pubmed, Embase, Google Scholar, Web of Science, and Cochrane CENTRAL. ** Two authors performed the screening.

**Table 1 jcm-13-05707-t001:** Study characteristics.

Study	Institution	Design	Number of Patients	Age (Years) Mean ± SD (Range)/Median	Postmenopausal Status, n (%)	Parity n (%) Mean/Median	Previous Hysterectomy n (%)	Previous Pelvic Surgery, n (%)	Previous POP Surgery, n (%)
Liu et al., 2018 [9]	Guangzhou Medical University, China	Retrospective	26	62.6 ± 7.3 (47–82)	n/a	>1 23 (100)	1 (4)	2 (8)	0
Lautherbach et al., 2020 [16]	Rambam Health Care Campus, Israel	Prospective	23	56.5 ± 7.7	n/a	n/a	0	0	0
Lowenstein et al., 2019 [8]	Rambam Health Care Campus, Israel AND Imelda Hospital, Belgium	Prospective	35	55 (40–81)	n/a	4 (1–6)	0	10 (28.5%)	n/a
Aharoni et al., 2021 [11]	Rambam Health Care Campus, Israel	Retrospective	65	59.93 ± 12.0	n/a	3.4 ± 1.7	0	14 (20)	n/a
Lu Z. et al., 2021 [13]	Hospital of Fudan University, China	Retrospective	35	53.7 ± 11.4	n/a	1.6 ± 0.8	1 (3)	n/a	1 (3)
Lu Z. et al., 2022 [12]	Hospital of Fudan University, China	Retrospective	55	54	35 (63.6)	=1 40 (72.7) =2 14 (25.5) =3 1 (1.8)	0	n/a	1 (1.8)
Farah et al., 2022 [4]	Lebanese American University Medical Center, Beirut	Prospective	23	56.7 ± 8.9	n/a	3 (2–3)	0	n/a	n/a
Wang X. 2022 [17]	Hospital of Fudan University, China	Prospective	15	60.67 (46–69)	12 (80)	1.23 (1–3)	0	1 (6.7)	n/a
Qin et al., 2022 [18]	Changzhou No.2 People’s Hospital, China	Prospective	18	62.61 ± 10.26	n/a	2.00 ± 0.77	n/a	n/a	0
Huang L. et al., 2022 [14]	Chengdu Women’s and Children’s Central Hospital, China	Retrospective	51	n/a	n/a	>1 51 (100)	n/a	n/a	n/a
Huang L. et al., 2023 [15]	Chengdu Women’s and Children’s Central Hospital, China	Retrospective	31	61.42 ± 8.969	24 (77.4)	3.81 ± 1.515	0	n/a	0
Ketenci Gencer et al., 2023 [7]	Istanbul Gaziosmanpasa Training and Research Hospital, Turkey	Prospective	37	57.7 ± 6.3	n/a	3.7 ± 1.8	2 (5.4)	5 (13.5)	3 (8.1)

n/a: not applicable.

**Table 2 jcm-13-05707-t002:** Operative data of the selected studies.

Study	N	Type of Surgery	Concomitant Anti-Incontinence Procedure, n (%)	Concomitant Anterior Colporrhaphy, n (%)	Concomitant Hysterectomy, n (%)	Concomitant Posterior Colporrhaphy, n (%)	Operative Time (min), Mean ± SD (Range)/Median	Blood Loss (mL), Mean ± SD (Range)/Median	Conversion Rate, n (%)	Hospital Stay (Days), Mean ± SD (Range) Median
Liu et al., 2018 [9]	26	SCP	0	0	25 (96)	0	189.74 ± 45.1	30.87 ± 20.8	3 (12) LPS	4.0 ± 1.6
Lautherbach et al., 2020 [16]	23	USLS	0	0	23 (100)	0	n/a	n/a	n/a	n/a
Lowenstein et al., 2019 [8]	35	USLS	4 (11) TVT-O	2 (6)	35 (100)	1 (3)	113 (89–233)	30 (20–200)		2 (1–3)
Aharoni et al., 2021 [11]	65	USLS	9 (13) TVT-O	13 (20)	65 (100)	12 (18)	141.4 ± 29.6	58 ± 68	0	3 (2–4)
Lu Z. et al., 2021 [13]	35	USLS	3 (8.5) MUS 2 (6) urethral folding	24 (69)	20 (57)	24 (69)	111.7 ± 39.4	67.9 ± 35.8	0	3.7 ± 1.1
Lu Z. et al., 2022 [12]	55	SCP	6 (10.9)	24 (43.6)	55 (100)	35 (63.6)	115.5 ± 18.4	86.2 ± 48.1	0	4.1 ± 1.4
Farah et al., 2022 [4]	23	USLS	0	20 (87)	23 (100)	20 (87)		85.2 ± 55.6	1 (4.3) vaginal	1.1 ± 0.3
Wang X. 2022 [17]	15	USLS+ ALLS	0	15 (100)	15 (100)	13 (86.7)	103.6 (65–166)	82 (50–200)	0	2 (2–4)
Qin et al., 2022 [17]	18	SSLS	0	n/a	14 (78)	n/a	192.78 ± 38.81	134.44 ± 111.21	0	7.94 (5–13)
Huang L. et al., 2022 [14]	51	12 (23) SCP 38 (74) SSLS 1 (2) USLS	n/a	n/a	n/a	n/a	59.58 ± 10.65 63.55 ± 9.12 52	147.50 ± 155.75 103.95 ± 138.22 100	0	5.58 ± 2.81 4.89 ± 1.18 7
Huang L. et al., 2023 [15]	31	SSLS	0	19 (61.3)	31 (100)	11 (35.5)	136.58 ± 37.39	82.52 ± 28.56	0	4.81 ± 1.25
Ketenci Gencer et al., 2023 [7]	37	LS	1 (2.7) TVT-O	3 (8.1)	18 (48.6)	n/a	60.3 ± 20.4	170.54 ± 117.11	3 (8.1) vaginal	n/a

Abbreviations: SCP = sacral colpopexy; USLS = uterosacral ligament suspension; ALLS = anterior longitudinal ligament suspension; SSLS = sacrospinous ligament suspension; LS = lateral suspension; TVT-O = transvaginal tape–obturatory; MUS = mid-urethral sling.

**Table 3 jcm-13-05707-t003:** Surgical outcomes of the selected studies.

Study	N	PreOP POP-Q Points Mean ± SD/Median Stage (Range) or n (%)	PostOP POP-Q Stage	De Novo SUI, n (%)	De Novo Constipation n (%)	De Novo Dyspareunia n (%)	Anatomical Success Rate (%)		Quality of Life Questionnaires		POP Recurrence n (%)	Reoperation Rate, n (%)	FUP (Months), Mean ± SD/Median (Range)
							Definition	Success Rate %	Type of Questionnaires—Preoperative Scores	Postoperative Scores			
Liu et al., 2018 [9]	26	≥stage 2 26 (100) Aa 1.4 ± 1.7 C 2.2 ± 1.9	Aa −1.85 ±0.6 C −6.1 ± 0.7	n/a	n/a	n/a	n/a	96	PFIQ-7 163.1 ± 46.2	18.4 ± 29.3	0	0	3 (3–14)
Lautherbach et al., 2020 [16]	23	3 (3–4)	0 (0–1)	n/a	n/a	n/a	n/a	n/a	FSFI Total 22.17 ± 1.62 Desire 3.87 ± 0.63 Arousal 3.72 ± 0.73 Lubrication 3.95 ± 0.66 Orgasm 3.02 ± 0.68 Satisfaction 3.44 ± 0.75 Pain 4.17 ± 0.25	28.66 ± 1.51 5.33 ± 0.52 5.48 ± 0.65 (*p* < 0.008) 4.46 ± 0.64 3.66 ± 0.63 5.42 ± 0.82 (*p* < 0.004) 4.31 ± 0.34	0	0	12
Lowenstein et al., 2019 [8]	35	Stage 3 (2–4) Stage II 4 (11) Stage III 25 (71) Stage IV 6 (17)	Stage 0–1 35 (100)	n/a	n/a	n/a	n/a	n/a	PFDI-20 Total 45 (26–54) POP subscale 53 (32–59)	6 (0–34) *p* < 0.005 2 (0–11) *p* < 0.005	0	0	3
Aharoni et al., 2021 [11]	65	Stage II 32 (49) Stage III 31 (48) Stage IV 2 (3)	n/a	0	n/a	n/a	n/a	n/a	n/a	n/a	n/a	0	n/a
Lu Z. et al., 2021 [13]	35	Stage III 33 (94.3) Stage IV 2 (5.7) Aa +0.6 ± 1.7 Ba +1.9 ± 2.2 C +1.5 ± 2.2 Ap −1.4 ± 1.0 Bp −1.1 ± 1.4 TVL +7.4 ±0.5	Aa −2.9 ± 0.2 Ba −2.9 ± 0.3 C −6.9 ± 0.9 Ap −3.0 ± 0.1 Bp −2.9 ± 0.1 TVL +7.2 ± 0.4	2 (6)	1 (3)	0	POP-Q < −1 cm	100	POPDI-6 9.9 ± 3.5 CRADI-8 2.5 ± 3.0 UDI-6 7.5 ± 4.4 Total PFDI-20 19.9 ± 6.7	0.9 ± 1.9 (*p* < 0.000) 0.7 ± 2.1 (*p* < 0.047) 1.6 ± 2.8 (*p* < 0.000) 3.2 ± 5.4 (*p* < 0.000)	0	0	3.9 ± 3.8 (1–13)
Lu Z. et al., 2022 [12]	55	Stage II 13 (23.6) Stage III 41 (74.5) Stage IV 1 (1.8) Aa 0.39 ±1.48 Ba 1.45 ±1.69 C 1.71 ± 2.52 p −1.75 ±1.13 Bp −1.14 ±1.74 TVL 6.87 ±1.39	Aa −2.86 ± 0.52 Ba −2.86 ±0.52 C −6.78 ±0.71 Ap −2.93 ± 0.33 Bp −2.93 ± 0.33 TVL 7.76 ± 0.72	n/a	n/a	n/a	POP-Q ≥ stage Il and any retreatment	52 (94.5)	n/a	n/a	3 (5.5)	0	35.5 ± 7.6 (24–46)
Farah et al., 2022 [4]	23	Stage I 1 (4) Stage II 6 (27) Stage III 15 (65) Stage IV 1 (4)	n/a	n/a	n/a	n/a	n/a	n/a	n/a	n/a	0	0	6 weeks
Wang X. 2022 [17]	15	Stage II 2 (13.3%) Stage III 13 (80%) Stage IV 1 (6.7%) Aa 1.07 ± 1.28. Ba 2.07 ± 1.58 C 2.40 ± 1.45 Ap 0.07 ± 1.39 Bp 0.26 ± 1.94 TVL 7.13 ± 0.35	Aa −2.93 ± 0.26 Ba −2.80 ± 0.41 C −7.13 ± 0.35 Ap −2.93 ± 0.26 Bp −2.87 ± 0.37 TVL 7.27 ± 0.46	n/a	n/a	n/a	n/a	n/a	POPDI-6 5.14 ± 3.37 CRADI-8 4.57 ± 3.65 UDI-6 5.57 ± 4.68 PFDI-20 18.50 ± 10.61	1.14 ± 1.17 (*p* < 0.000) 1.5 ± 1.29 (*p* < 0.001) 1.14 ± 1.46 (*p* < 0.000) 3.79 ± 2.55 (*p* < 0.000)	0	0	9.93 (9–12)
Qin et al., 2022 [18]	18	Aa 1.00 ± 1.00 Ba 2.31 ± 1.19 C 5.19 ± 2.18 Ap 0.86 ± 0.97 Bp 1.64 ± 1.08 TVL 8.72 ± 0.46	Aa −2.17 ± 0.45 Ba −2.52 ± 0.17 C −7 ± 0.41 Ap −2.32 ± 0.20 Bp −2.55 ± 0.23 TVL 7.31 ± 0.57	n/a	n/a	n/a	n/a	n/a	n/a	n/a	0	0	6
Huang L. et al., 2022 [14]	51	n/a	n/a	n/a	n/a	n/a	n/a	n/a	n/a	n/a	n/a	n/a	n/a
Huang L. et al., 2023 [15]	31	Stage I 2 (6.5) Stage II 7 (22.6) Stage III 21 (67.7) Stage IV 1 (3.2)	n/a	n/a	n/a	n/a	POP-Q stage ≤ I	74.2	n/a	PFIQ-7 23.20 ± 8.32 UIQ-7 11.06 ± 6.55 POPIQ-7 5.99 ± 2.45 CRAIQ-7 6.14 ± 3.06 PFDI-20 47.88 ± 11.67 UDI-6 19.35 ± 4.51 POPDI-6 13.31 ± 4.22 CRADI-8 15.22 ± 4.25	8 (25.8)	n/a	12
Ketenci Gencer et al., 2023 [7]	37	Stage ≥ 3 37 (100) Aa 2.17 ± 0.67 Ba 1.88 ± 0.87 Ap 1.37 ± 0.61 Bp 1.58 ± 1.17 C 3.83 ± 1.0003 D 3 ± 1.13 TVL 7.35 ± 0.99	Aa −1.32 ± 1.66 Ba −0.75 ± 2.246 Ap −1.71 ± 1.5 Bp −1.41 ± 2.11 C −5.78 ± 4.18 D −6.35 ± 3.54 TVL 7.56 ± 1.32	1 (2.7)	5 (13.5)	0	n/a	n/a	PISQ-12 Behavioral factor 5.94 ± 1.99 Physical factor 2.72 ± 2.16 Partner-related factor 4.54 ± 2.34 Total 13.13 ± 3.6	9.59 ± 1.88 8.67 ± 4.74 7.48 ± 3.02 25.89 ± 5.96 *p* value <0.001 for all items	5 (13.5)	5 (13.5)	6

**Table 4 jcm-13-05707-t004:** Complications of the selected studies.

Study	N	Overall Complication Rate n, (%)	Intraoperative Complication Rate, n (%) Satava Classification			Postoperative Complication Rate, n (%) Clavien–Dindo Classification					
			1	2	3	1	2	3a	3b	4	5
Liu et al. 2018 [9]	26	0	0	0	0	0	0	0	0	0	0
Lautherbach et al., 2020 [16]	23	0	0	0	0	0	0	0	0	0	0
Lowenstein et al., 2019 [8]	35	1 (3) postoperative	0	0	0	0	1 (3) ileus treated conservatively	0	0	0	0
Aharoni et al., 2021 [11]	65	4 (6) intraoperative 2 (3) postoperative	1 (1) intra-abdominal bleeding	1 (1) vaginal bleeding 2 (3) cystotomy	0	0	1 (1) infection, 1 (1) hematoma	0	0	0	0
Lu Z. et al., 2021 [13]	35	0	0	0	0	1 (1) urinary retention	1 (3) infection	0	0	0	0
Lu Z. et al., 2022 [12]	55	5 (9) postoperative 3 (5.5) mesh exposure after 1 year	0	0	0	2 (3.6) urinary retention	2 (3.6) urinary tract infection, 1 (1.8) hematoma	0	0	0	0
Farah et al., 2022 [4]	23	1 (4) intraoperative 1 (4) postoperative	1 (4.3) ureteral kinging	0	0	1 (4) urinary retention	0	0	0	0	0
Wang X. 2022 [17]	15	0	0	0	0	0	0	0	0	0	0
Qin et al., 2022 [18]	18	2 (11) postoperative	0	0	0	0	1 (5) dull pain, 1 (5) lumbosacral swelling	0	0	0	0
Huang L. et al., 2022 [14]	51	10 (19.6) postoperative	n/a	n/a	n/a	n/a	n/a	n/a	1 (8.3) * mesh exposure	n/a	n/a
Huang L. et al., 2023 [15]	31	5 (16.1) postoperative 23 (74.2) BUTTOCK PAIN REPORTED at 1 year	0	0	0	5 (16.1) not specified	0	0	0	0	0
Ketenci Gencer et al., 2023 [7]	37	20 (54.1)	0	3 (8.1) bladder injury 1 (2.7) epigastric vessel injury	0	0	1 (2.7) pelvic pain, 5 (13.5) constipation, 3 (5.4) urinary tract infection, 3 (8.1) insertion pain, 2 (5.4) vaginal hematoma, 1 (2.7) mesh erosion treat with estrogens, 2 (5.4) vaginitis	0	1 (2.7) mesh erosion surgically removed	0	0

* considering 12 sacral colpopexy performed.

**Table 5 jcm-13-05707-t005:** Risk of bias according to ROBINS-I.

Study	Bias Due to Confounding	Bias in Selection of Participants into the Study	Bias in Classification of Interventions	Bias Due to Deviations from Intended Intervention	Bias Due to Missing Data	Bias in Measurement of Outcomes	Bias in Selection of the Reported Results	Overall Bias
Liu et al., 2018 [9]								
Lautherbach et al., 2020 [16]								
Lowenstein et al., 2019 [8]								
Aharoni et al., 2021 [11]								
Lu Z. et al., 2021 [13]								
Lu Z. et al., 2022 [12]								
Farah et al., 2022 [4]								
Wang X. 2022 [17]								
Qin et al., 2022 [18]								
Huang L. et al., 2022 [14]								
Huang L. et al., 2023 [15]								
Ketenci Gencer et al., 2023 [7]								

Green color indicates low possibility of bias, yellow color indicates moderate possibility of bias, red color indicates serious possibility of bias.

## Data Availability

Data will be made available upon request.

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
