# Peer review of "Transvaginal Natural Orifice Transluminal Endoscopic Surgery (vNOTES) in Urogynecological Surgery: A Systematic Review"

_jcm, 2024, doi:10.3390/jcm13195707_

Round 1

Reviewer 1 Report

Comments and Suggestions for Authors

The paper provides a comprehensive review of transvaginal natural orifice transluminal endoscopic surgery (vNOTES) in urogynaecological procedures. It systematically evaluates the efficacy, safety, and feasibility of vNOTES, highlighting its potential benefits such as improved cosmetic outcomes, reduced surgical injury, less postoperative pain, and quicker patient discharge. The review includes 12 studies, focusing on various procedures like uterosacral ligament suspension, sacral colpopexy, and sacrospinous ligament suspension, reporting high success rates and low complication rates.

Highlights of the current manuscript can be listed as:

  • The paper covers a wide range of studies and procedures, providing a thorough overview of the current state of vNOTES in urogynecology.
  • The included studies report high success rates (>90%) for the procedures, indicating the potential effectiveness of vNOTES.
  • The review highlights the low incidence of complications, which supports the safety of vNOTES.
  • The paper follows a systematic approach, including PROSPERO registration and adherence to PRISMA guidelines, ensuring the reliability of the review process.
  • The paper identifies the need for further randomized clinical trials, providing a clear direction for future research.

However, the paper can be improved on the number of fronts:

  • The definitions of success varied across the included studies, making it difficult to compare outcomes directly.
  • The risk of bias in the included studies was rated as medium to high, which could affect the reliability of the findings.
  • Most studies had short follow-up periods, limiting the ability to assess long-term outcomes and complications.
  • The review did not include any randomised controlled trials, which are considered the gold standard for clinical research.
  • The paper acknowledges the steep learning curve associated with vNOTES, which may limit its widespread adoption.

Overall, the paper provides valuable insights into the potential benefits and challenges of vNOTES in urogynaecological surgery, while also highlighting areas where further research is needed.

Comments on the Quality of English Language

proof-reading requiered

Author Response

The paper provides a comprehensive review of transvaginal natural orifice transluminal endoscopic surgery (vNOTES) in urogynaecological procedures. It systematically evaluates the efficacy, safety, and feasibility of vNOTES, highlighting its potential benefits such as improved cosmetic outcomes, reduced surgical injury, less postoperative pain, and quicker patient discharge. The review includes 12 studies, focusing on various procedures like uterosacral ligament suspension, sacral colpopexy, and sacrospinous ligament suspension, reporting high success rates and low complication rates.

Highlights of the current manuscript can be listed as:

  • The paper covers a wide range of studies and procedures, providing a thorough overview of the current state of vNOTES in urogynecology.
  • The included studies report high success rates (>90%) for the procedures, indicating the potential effectiveness of vNOTES.
  • The review highlights the low incidence of complications, which supports the safety of vNOTES.
  • The paper follows a systematic approach, including PROSPERO registration and adherence to PRISMA guidelines, ensuring the reliability of the review process.
  • The paper identifies the need for further randomized clinical trials, providing a clear direction for future research.

Answer: We thank the reviewer for the positive comments. 

However, the paper can be improved on the number of fronts:

  • The definitions of success varied across the included studies, making it difficult to compare outcomes directly.
  • The risk of bias in the included studies was rated as medium to high, which could affect the reliability of the findings.
  • Most studies had short follow-up periods, limiting the ability to assess long-term outcomes and complications.
  • The review did not include any randomised controlled trials, which are considered the gold standard for clinical research.
  • The paper acknowledges the steep learning curve associated with vNOTES, which may limit its widespread adoption.

Answer: We thank the reviewer for his/her comment and for the possibility to improve our manuscript. The reviewer clearly highlights the limitations of our review which clearly depend on the available evidence. Although we can only hope for better and more homogeneous evidence in the future  to improve this review, we have better underlined these limitations in the limitations section:

See Limitations section

‘The limitations of this review include the variation in definitions of success across the included studies, making direct comparisons of outcomes challenging. The risk of bias in the studies was rated as medium to high, potentially affecting the reliability of the findings. Additionally, most studies had short follow-up periods, limiting the assessment of long-term outcomes and complications. The review did not include any randomized controlled trials, which are the gold standard in clinical research. Furthermore, the steep learning curve associated with vNOTES may hinder its broader adoption in clinical practice. However, notwithstanding these limitations, the present review provides a clear picture of the current state of the art of vNOTES surgery.’

Reviewer 2 Report

Comments and Suggestions for Authors

Your study is about vNOTES technique and Urogynaecology. Congratulations on your interest about endoscopic surgery, a continuously improving field.

Regarding evidence aquisition part, the material and methods should be more clearly described - especialy inclusion and excludion criteria.

Regarding discussion section - all advantages associated with this technique should be moved to introduction.

In coclusion section you can include a take home message for clinicians specialised in urogynaecology - eg - importance of case selection.

Author Response

Your study is about vNOTES technique and Urogynaecology. Congratulations on your interest about endoscopic surgery, a continuously improving field.

We thank the reviewer for taking time to review our manuscript.

Regarding evidence acquisition part, the material and methods should be more clearly described - especialy inclusion and excludion criteria.

We thank the reviewer for his/her comment and for the possibility to improve our manuscript. Following the reviewer recommendations, the methods section has been improved.

Studies were selected using the following PICO protocol. Patients: women undergoing urogynaecology surgery; Intervention: vNOTES; Comparator: any other urogynaecology intervention; Outcome: success. We included cohort and case-control studies, as well as non-randomised (NRCT) or randomised trials (RCT).

Exclusion criteria included: manuscripts reporting <10 cases or written in languages other than English were excluded, as were review articles, video articles, or articles not inherent to the topic.

Regarding discussion section - all advantages associated with this technique should be moved to introduction.

We thank the reviewer for his/her comment. We appreciate his comment, and we agree that it may be use-full to add some advantages in the introduction section. The section on advantages as it is in the discussion section is too long to fit the introduction section which should be typically shorter and straightforward. In order to follow the reviewer suggestion, we have summarized some advantages of vNOTES in the introduction section.

See Introduction section: 

‘From a technical perspective, vNOTES offers several advantages over trans-abdominal LESS, such as a larger and more flexible colpotomy opening, reduced instrument clashing, and better visualization due to the "bottom-up" camera positioning and smoke plume dispersion. The vNOTES approach also reduces the need for extensive adhesiolysis in patients with severe abdominal adhesions and provides better maneuverability and control of the uterine blood supply during procedures for enlarged uteri. In pelvic organ prolapse (POP) surgery, vNOTES offers improved visualization for precise suturing, easier adnexectomy, and significant cosmetic benefits due to the absence of visible laparoscopic scars.’

In coclusion section you can include a take home message for clinicians specialised in urogynaecology - eg - importance of case selection.

We thank the reviewer for his/her comment and for the possibility to improve our manuscript. A take home message has been included as suggested.

See Conclusion section: 

'Clinicians beginning to perform vNOTES surgery should proceed carefully, ensuring adequate mentorship and careful case selection to maximize outcomes. Meanwhile, clinicians who are experts in vNOTES should focus on designing well-structured clinical trials to better define the role of vNOTES in urogynaecological surgery.'

Reviewer 3 Report

Comments and Suggestions for Authors

Manuscript ID - jcm-3170698

Review

Title: Transvaginal natural orifice transluminal endoscopic surgery (V-NOTES) in urogynecological surgery: a systematic review

I had a pleasure reviewing systematic review concerning Transvaginal Natural Orifice Transluminal Endoscopic Surgery (V-NOTES) in which you evaluated 12 manuscripts concerning this topic.

Minimal invasive surgery is becoming more actual, especially regarding the cost-benefit. At the same time, indications for vaginal surgery, which is, also, minimally invasive, has been based on the problems concerning prolapse and statics of female genital organs. vNOTES is not a new surgical technique, but rather a new surgical approach that combines and modifies several existing minimally invasive techniques. Therefore, a study reviewing published data reporting V-NOTES is welcomed.

Introduction describes V-NOTES in detail, and Evidence Acquisition and Evidence Synthesis clearly describe methodology and results, including Bias Assessment. 602 publications were identified, and only 12 included in the study. Eliminating process was written, but I would suggest the authors to insert a flow chart diagram, presenting us the selection of the studies included in the review. Discussion and Conclusion are well written, emphasizing the limitations of the study. References are correct.

Therefore, I would suggest this study with minimal corrections.

Author Response

Review

Title: Transvaginal natural orifice transluminal endoscopic surgery (V-NOTES) in urogynecological surgery: a systematic review

I had a pleasure reviewing systematic review concerning Transvaginal Natural Orifice Transluminal Endoscopic Surgery (V-NOTES) in which you evaluated 12 manuscripts concerning this topic.

Minimal invasive surgery is becoming more actual, especially regarding the cost-benefit. At the same time, indications for vaginal surgery, which is, also, minimally invasive, has been based on the problems concerning prolapse and statics of female genital organs. vNOTES is not a new surgical technique, but rather a new surgical approach that combines and modifies several existing minimally invasive techniques. Therefore, a study reviewing published data reporting V-NOTES is welcomed.

Introduction describes V-NOTES in detail, and Evidence Acquisition and Evidence Synthesis clearly describe methodology and results, including Bias Assessment. 602 publications were identified, and only 12 included in the study. Eliminating process was written, but I would suggest the authors to insert a flow chart diagram, presenting us the selection of the studies included in the review. Discussion and Conclusion are well written, emphasizing the limitations of the study. References are correct.

Therefore, I would suggest this study with minimal corrections.

Answer: 

We thank the reviewer for the positive comments, for taking time to review our manuscript and for the suggestions. 

The reviewer can find the PRISMA flow chart in the supplementary materials of the article. 

Reviewer 4 Report

Comments and Suggestions for Authors

The study "Transvaginal Natural Orifice Transluminal Endoscopic Surgery (V-NOTES) in Urogynecological Surgery: A Systematic Review" is a very interesting work analyzing the efficacy, safety, and feasibility of this technique for surgical treatment of pelvic organ prolapse. It describes in a clear language all the advances of this type of approach by using official methods for conduction of systematic review studies.

The flow of the story and the sections/sub-sections of the manusript are excellent and easy to follow.

Conclusions are in accordance with the results of the analyzed studies.

There are only a couple of points to pay attention on, as follows:

Line 45 - Please consider re-positioning reference number 2 after "over the past few years".

Lines 181-191 (sub-section 3.3.2. Anatomical and Functional Outcomes) - Please consider describing in a little more detail how anatomical outcomes were defiend and assessed in analyzed studies. Is it assessed on any way other than POP-Q system?

Line 221 - Please consider rephrasing the phrase "less bloody".

Why no limits on time or type of study were applied? Please consider mentioning it as one of the potential limitations of this study.

If possible, please describe what the analyzed studies say about controling infection during and after vNOTES procedures?

Please consider adding a couple of additional important findings in the Conclusion section.

Author Response

The study "Transvaginal Natural Orifice Transluminal Endoscopic Surgery (V-NOTES) in Urogynecological Surgery: A Systematic Review" is a very interesting work analyzing the efficacy, safety, and feasibility of this technique for surgical treatment of pelvic organ prolapse. It describes in a clear language all the advances of this type of approach by using official methods for conduction of systematic review studies.

The flow of the story and the sections/sub-sections of the manusript are excellent and easy to follow.

Conclusions are in accordance with the results of the analyzed studies.

 We thank the reviewer for the positive comments.

There are only a couple of points to pay attention on, as follows:

Line 45 - Please consider re-positioning reference number 2 after "over the past few years".

We thank the reviewer for the comment. We have modified the manuscript as suggested.

Lines 181-191 (sub-section 3.3.2. Anatomical and Functional Outcomes) - Please consider describing in a little more detail how anatomical outcomes were defiend and assessed in analyzed studies. Is it assessed on any way other than POP-Q system?

We thank the reviewer for the comment. Table 3 describes in detail the anatomical and functional outcomes assessed by each study. Overall, most of the studies used the POP-Q system (11/12) however other outcomes included anatomical success and patient reported outcomes assessed with specific questionnaires. Moreover recurrence and reoperation rates were assessed. We have improved the manuscript as follows:

‘Overall, most of the included studies (11 out of 12) used the POP-Q system to evaluate anatomical outcomes. Additionally, anatomical success rates and various patient-reported outcomes were assessed using dedicated questionnaires (PFIQ, FSFI, PFDI, POPDI, PISQ). Finally, recurrence and reoperation rates were also reported. Table 3 provides a detailed description of the anatomical and functional outcomes for each study.

Line 221 - Please consider rephrasing the phrase "less bloody".

We thank the reviewer for the comment. We have modified the manuscript as suggested.

See new phrase:

‘Even though most of the vNOTES POP reconstructive procedures were associated with hysterectomy, overall estimated blood loss (EBL) was superior to that registered after a conventional laparoscopic approach27–29, but when compared with conventional vaginal procedures, vNOTES appeared to be a safer in terms of blood loss.

Why no limits on time or type of study were applied? Please consider mentioning it as one of the potential limitations of this study.

We thank the reviewer for the comment. The lack of limitations on study type or time depends on the little evidence available on the subject. Although we agree it may be considered a limitation, we preferred to collect all the available evidence on the subject to give an overview of the actual of the art of the technique.

Limitations section has been updated as suggested:

The absence of restrictions on time and study type may be considered a limitation of this study. However, due to the limited number of studies available on the subject, we chose to take a broader approach to provide a comprehensive overview of the current state of the art of the technique.

If possible, please describe what the analyzed studies say about controling infection during and after vNOTES procedures?

We thank the reviewer for his/her comments. In the discussion section we have added a paragraph describing these aspects.

In terms of infection control the selected studies applied different strategies. More specifically, antibiotic prophylaxis with a cephalosporin alone or in combination with metronidazole was performed preoperatively. In case of postoperative antibiotic prophylaxis, the cephalosporin was stopped within 48 hours after the surgery was completed. Overall, the rate of infections using these strategies was low (2,2 %).

Please consider adding a couple of additional important findings in the Conclusion section.

We thank the reviewer for his/her suggestions. In accordance with another reviewers suggestions. Conclusion section has been improved as follows.

According to our systematic review, vNOTES surgery appears to be a safe and effective approach for patients with pelvic organ prolapse. Moreover, the results identify USLS as a safe and effective technique for the suspension of the apex in POP. However, further prospective randomised studies with a longer follow-up are needed to confirm the available data. Clinicians beginning to perform vNOTES surgery should proceed carefully, ensuring adequate mentorship and careful case selection to maximize outcomes. Meanwhile, clinicians who are experts in vNOTES should focus on designing well-structured clinical trials to better define the role of vNOTES in uro-gynecological surgery.